# The Quadratic Constitutive Model Based on Partial Derivative and Taylor Series of Ti6242s Alloy and Predictability Analysis

**DOI:** 10.3390/ma16072928

**Published:** 2023-04-06

**Authors:** Jiansheng Zhang, Guiqian Xiao, Guoyong Deng, Yancheng Zhang, Jie Zhou

**Affiliations:** 1Chongqing Key Laboratory of Advanced Mold Intelligent Manufacturing, College of Materials Science and Engineering, Chongqing University, Chongqing 400044, China; zhangjiansheng@cqu.edu.cn (J.Z.); xgq3790@163.com (G.X.); 202109131199@cqu.edu.cn (G.D.); m13983116464@163.com (Y.Z.); 2Chongqing Jiepin Technology Co., Ltd., Chongqing 400000, China

**Keywords:** Taylor series, partial derivative, hot compression, prediction accuracy, quadratic model

## Abstract

To solve the problem of insufficient predictability in the classical models for the Ti6242s alloy, a new constitutive model was proposed, based on the partial derivatives from experimental data and the Taylor series. Firstly, hot compression experiments on the Ti6242s alloy at different temperatures and different strain rates were carried out, and the Arrhenius model and Hensel–Spittel model were constructed. Secondly, the partial derivatives of logarithmic stress with respect to temperature and logarithmic strain rate at low, medium and high strain levels were analyzed. Thirdly, two new constitutive models with first- and second-order approximation were proposed to meet the requirements of high precision. In this new model, by analyzing the high-order differential data of experimental data and combining the Taylor series theory, the minimum number of terms that can accurately approximate the experimental rheological data was found, thereby achieving an accurate prediction of flow stress with minimal material parameters. In the new model, by analyzing the high-order differential of the experimental data and combining the theory of the Taylor series, the minimum number of terms that can accurately approximate the experimental rheological data was found, thereby achieving an accurate prediction of flow stress with minimal material parameters. Finally, the prediction accuracies for the classical model and the new model were compared, and the predictabilities for the classical models and the new model were proved by mathematical means. The results show that the prediction accuracies of the Arrhenius model and the Hensel–Spittel model are low in the single-phase region and high in the two-phase region. In addition, second-order approximation is required between the logarithmic stress and logarithmic strain rate, and first-order approximation is required between logarithmic stress and temperature to establish a high-precision model. The order of prediction accuracy of the four models from high to low is the quadratic model, Arrhenius model, linear model and HS model. The prediction accuracy of the quadratic model in all temperatures and strain rates had no significant difference, and was higher than the other models. The quadratic model can greatly improve prediction accuracy without significantly increasing the material parameters.

## 1. Introduction

The Ti6242s alloy, a typical high-temperature titanium alloy, is widely used in high-pressure compressor disks and blades in aeroengines; these parts are usually manufactured by a hot-working process [1,2], such as hot forging, hot extrusion, hot roll forging, etc. The most important purpose of hot-working processes is that metals are deformed into the desired shapes under certain thermomechanical processing conditions to obtain the appropriate microstructure and mechanical performance [3]. Numerical simulation technology is an important technical means for achieving the accurate control of shape and performance, and one of the cores of accurate numerical simulation is to build a high-precision constitutive model [4].

In past decades, many scholars have studied the constitutive model of titanium alloy. Three main categories of constitutive models are utilized to predict the flow behavior of metallic alloys, which are based on physical, phenomenological and artificial neural network constitutive models [5,6,7,8,9]. The most widely used constitutive models in numerical simulation software are the Arrhenius (AH) model [10,11], the Johnson-Cook (JC) model [12], the Hensel–Spittel (HS) model [13,14] and so on. Since these constitutive models were proposed, many subsequent studies have attempted to improve these classical models. For example, Kotkunde et al. [15] used four constitutive models (a modified JC model, modified AH model, modified Zerilli–Armstrong and the Rusinek–Klepaczko model) to analyze the accuracy of the rheological behavior of the Ti-6Al-4V alloy. The results showed that the modified AH model had high accuracy. Zhang et al. [16] proposed the strain-compensated AH and modified Norton–Hoff constitutive models for the Ti-6Al-4V alloy. The results showed that the Norton–Hoff model had higher accuracy. Xiao et al. [17] proposed a new constitutive model of the TiNiNb alloy, in which the effects of temperature and strain rate on stress were considered. The regression results of the experimental data showed that the predicted accuracy of the proposed constitutive model was very high. Luo et al. [18] analyzed the Kocks–Mecking constitutive model of the Ti-6Al-4V alloy, and the average relative error was about 6.13%. Cai et al. [19] conducted a constitutive analysis of the Ti-6Al-4V alloy by using the stress–strain data obtained from an isothermal hot compression experiment. Considering the independent effects of strain, strain rate and temperature, a modified Arrhenius constitutive model was proposed. The results showed that the modified parallel constitutive model based on multiple regression could better predict the flow stress of the Ti-6Al-4V alloy, and had good correlation and generalization ability. Ming et al. [20] compared the prediction errors of the modified AH model and the modified JC model for the Ti-6Al-4V alloy. The results showed that the modified AH model with strain compensation was more accurate than the modified JC model. Ga et al. [21] compared the prediction accuracy of five constitutive models for the flow data of the Ti-6Al-4V alloy. The results showed that the modified HS model had the highest accuracy and the JC model had the worst accuracy. Recently, an artificial neural network model was also introduced into the prediction of rheological data. For example, Ahmed et al. [22] compared the prediction accuracy between the AH model and an artificial neural network (ANN) model. The results showed that the predictability of the artificial neural network model was higher than that of the AH model for the Ti-2.5Al-1.8Mn alloy. Sun et al. [23] proposed a BP neural network constitutive model of the Ti40 alloy, and the results showed that this model had a high prediction accuracy for flow stress. Reddy et al. [24] established a BP neural network constitutive model to predict the flow stress of the Ti-6Al-4V alloy. This model was successfully trained by using experimental flow data in both the double phase α + β region and the single phase (β) region. This model seemed to have a higher prediction accuracy than the classical models. However, there are many obvious disadvantages for the neural network model, such as many offset and weight parameters, no gradient information (simple expression), over fitting, etc. In addition, compared with the traditional models, the calculation efficiency of the artificial neural network model is extremely low. For finite element simulation with a strict speed requirement, the artificial neural network model will obviously not be adopted by the mainstream commercial simulation software. Because the AH, JC and HS models have the characteristics of simple structure and high accuracy, they are currently the mainstream models.

However, it is difficult for classical models to accurately predict the flow stress in both the α + β phase and β phase regions due to the phase transition of the Ti6242s alloy within the hot forming temperature range. In addition, the second-order approximation is required between the logarithmic stress and logarithmic strain rate, and first-order approximation is required between logarithmic stress and temperature to establish a high-precision model. However, the classical constitutive models have only first-order accuracy; therefore, these modifications and enhancements conducted by the above researchers have limited improvement in the prediction accuracy, only within the range of the first-order accuracy.

To solve the problem of insufficient predictability in the classical constitutive models, a novel constitutive model of the Ti6242 alloy was established based on the partial derivative and Taylor series. The novel constitutive model can predict the high-temperature flow stress of the Ti6242s alloy in both the α + β phase and single β phase regions more precisely. Firstly, the isothermal hot compression tests were completed in the temperature range of 900~1100 °C and a strain rate range of 0.001~10 s^–1^ on a Gleeble-3500 simulator. Secondly, two new constitutive models based on the Taylor series were proposed without significantly increasing the material parameters. In this novel building method, the minimum number of terms that can accurately approximate the experimental rheological data was found by analyzing the high-order differentials of experimental data and combining the theory of the Taylor series, thereby achieving an accurate prediction of flow stress with minimal material parameters. Finally, the prediction accuracy of the proposed constitutive equation was evaluated by comparing with classical constitutive models. The results showed that the prediction accuracy of the classical models was low in the single-phase region and high in the two-phase region. The order of prediction accuracy of the four models from high to low is quadratic model, Arrhenius model, linear model and HS model. The prediction accuracy of the quadratic model in all temperatures and strain rates is higher than the other models and it can greatly improve the prediction accuracy without significantly increasing the material parameters.

## 2. Materials and Experiments

### 2.1. Material

The experimental material was the Ti6242s alloy and its nominal chemical composition is shown in Table 1. The main alloy elements of the experimental material were Al, Sn, Zr and Mo. It is a kind of near-α titanium alloy with a service temperature between 450 °C and 500 °C. This material is mainly used for medium and high pressure compressor discs and blades in aircraft engines [25]. It has excellent high temperature durability, weldability and processability.

Figure 1 shows the original forged microstructure of the material. It can be seen that the primary microstructure of the material displays the typical bimodal morphology which is mainly composed of massive equiaxed primary α phase (more than 50%), lamellar secondary α phase and a small amount of transformed phase β.

The hot processing or heat treatment of the titanium alloy is carried out in a two-phase zone or a single-phase zone, which has a certain impact on the structure and properties. Therefore, the phase transformation point of the titanium alloy is an important parameter in its processing specifications. For the Ti6242s alloy, the phase transform temperature from α to β is 1015 °C.

### 2.2. Experimental Details and Results

Generally, the range of the forming temperature of titanium alloy is located in a double phase region [26,27,28]. Therefore, in the isothermal hot compression, the temperatures located in both the double phase region (900, 950, 1000 °C) and single phase region (1050 and 1100 °C) were selected to build a complete database. Twenty-five experimental specimens were machined from the original metal. The size of the specimens was 8 mm in diameter and 12 mm in height. As shown in Table 2, 25 samples were heated to 900, 950, 1000, 1050 and 1100 °C at the rate of 5 °C/s and then subjected to isothermal compression on the Geeble-3500 thermal simulation testing machine. The compression deformation rates were 0.001, 0.01, 0.1, 1 and 10 s^−1^, respectively, and the compression amounts were 60% (the true strain was 0.916). To retain the microstructure, the specimens were immediately placed in water. To eliminate the friction, tantalum foil with a thickness of 0.1 mm was placed between die and specimen before hot compression.

Figure 2 shows the stress–strain curves at various strain rates and temperatures. The following rules can be obtained from Figure 2: (1) the flow stress decreases with the increase in temperature under the same strain rate. The higher the temperature is, the stronger the atomic activity and the stronger the dislocation motion ability are, and the speed of mutual cancellation between dislocations is accelerated. Meanwhile, with the increase in temperature, α→β phase transformation is triggered. The slip system of β phase is more than α phase. Then, this phase transformation causes deformation to occur more easily. In addition, the increase in temperature accelerates the nucleation and growth rate of recrystallization, and the softening effect is enhanced. (2) In the temperature range of both α + β phase region (900~1000 °C), a significant stress drop can be observed at all strain rates, and the flow softening phenomenon is more obvious when the temperature lowers. (3) In single β phase zone (1050~1100 °C), the curve will soften only when the strain rate is low, and the curve will decline slightly or remain basically unchanged after reaching the peak value at the high strain rate. The reason is that the temperature rises above the recrystallization termination temperature of the material; recrystallization stops in the grains, and the softening effect of dynamic recovery is not obvious; under the counteraction of deformation hardening, it shows relatively stable flow stress.

### 2.3. Strain Dispersion

In order to regress the constitutive model easily, the flow stress curve was divided into 10 equal parts according to the strain from 0.04 to 0.9. It is worth noting that the accuracy increases with more divided parts in a certain range. In order to eliminate the influence of the number of equal parts on the accuracy of the model, all subsequent models used the same number of equal parts, that is, the data in Table 3. The data in Table 3 can be obtained by linear interpolation using each rheological curve in Figure 2.

## 3. Classical Constitutive Model

The accurate construction of the prediction model of the flow curve is the premise of accurate numerical simulation. Before proposing a new phenomenological constitutive model, two classical constitutive models, namely the Arrhenius model and the Hensel–Spittel (HS) model, were analyzed.

### 3.1. Arrhenius Model

The Arrhenius constitutive model has been widely used due to its high accuracy and wide generalizability. The model was first proposed by Sellars and McTegart [29], and its specific form is presented in Equation (1).
(1)$$\{\begin{array}{l} \dot{\varepsilon }=A\sigma^{\frac{\beta }{\alpha }}\exp\left(-\frac{Q}{RT}\right),\;\left(\alpha \sigma \leq 0.8\right)\\ \dot{\varepsilon }=A\exp\left(\beta \sigma \right)\exp\left(-\frac{Q}{RT}\right),\left(\alpha \sigma \geq 1.2\right)\\ \dot{\varepsilon }=A{\left[\sinh\left(\alpha \sigma \right)\right]}^{n}\exp\left(-\frac{Q}{RT}\right),\;\left(\mathrm{for}\;\mathrm{all}\right) \end{array}$$
where *A*, *α*, *n* and *β* are material parameters, *Q* is the activation energy of thermal deformation (J·mol^−1^), *R* is the universal gas constant (8.314 J·K^−1^·mol^−1^), T is the absolute temperature (K), $\dot{\varepsilon }$ is the strain rate (s^−1^), *σ* is flow stress (MPa). In Equation (1), the effect of strain on flow stress was not considered. In recent years, most scholars consider how to take the effect of strain into account [20]. Equation (2) can be obtained by logarithmic transform in Equation (1).
(2)$$\{\begin{array}{l} \ln\dot{\varepsilon }=\ln A+\frac{\beta }{\alpha }\ln\sigma -\frac{Q}{RT},\left(\alpha \sigma \leq 0.8\right)\\ \ln\dot{\varepsilon }=\ln A+\beta \sigma -\frac{Q}{RT},\left(\alpha \sigma \geq 1.2\right)\\ \ln\dot{\varepsilon }=\ln A+n\ln\sinh\left(\alpha \sigma \right)-\frac{Q}{RT},\left(\mathrm{for}\;\mathrm{all}\right) \end{array}$$

The material parameters of lnA, α, n, Q at different strain levels can be obtained by multivariate nonlinear regression of equation $\ln\dot{\varepsilon }=\ln A+n\ln\sinh\left(\alpha \sigma \right)-Q/\left(RT\right)$ using flow stress data (shown in Figure 2). For example, the strain has been divided into 10 parts between 0.04 and 0.9. Then, the stress matrix (shown in Table 3), corresponding to different strain levels, was obtained by interpolation. The material parameters of lnA, α, n and *Q* corresponding to different strain levels can be obtained by multivariate nonlinear regression using the stress matrix and equation $\ln\dot{\varepsilon }=\ln A+n\ln\sinh\left(\alpha \sigma \right)-Q/\left(RT\right)$. Then, the polynomial fittings of lnA-ε, α-ε, n-ε and Q-ε curves (shown in Figure 3) were performed, respectively, to obtain the constitutive equation with strain compensation.

As shown in Figure 3, the ten points in each subgraph correspond to strains of 0.04, 0.14, 0.23, …, 0.90. The fifth-degree polynomial is used to fit these data, and expression of each parameters is obtained. The degree of polynomial can be determined according to the regression accuracy, but too much will lead to overfitting.
(3)$$\{\begin{array}{l} \ln A=43.91+192.62\varepsilon -969.19\varepsilon^{2}+2137.32\varepsilon^{3}-2236.13\varepsilon^{4}+887.20\varepsilon^{5}\\ \alpha =0.01-0.05\varepsilon +0.22\varepsilon^{2}-0.45\varepsilon^{3}+0.47\varepsilon^{4}-0.19\varepsilon^{5}\\ n=4.61-3.33\varepsilon -18.75\varepsilon^{2}+46.38\varepsilon^{3}-57.78\varepsilon^{4}+26.69\varepsilon^{5}\\ Q=491161+1703450\varepsilon -9111853\varepsilon^{2}+20591837\varepsilon^{3}-21622908\varepsilon^{4}+8542954\varepsilon^{5} \end{array}$$

The Arrhenius constitutive equation of the Ti6242s alloy can be obtained by introducing Formula (3) into (1). In order to observe the prediction accuracy of the model more intuitively, the experimental data and the prediction data of the Arrhenius constitutive equation were plotted in the same figure.

As shown in Figure 4, the prediction accuracy of the Arrhenius constitutive equation is significantly different in different temperatures and strain rates. The prediction accuracy of the Arrhenius constitutive equation is low in the single-phase region (900, 950 and 100 °C) and high in the two-phase region (1050 and 1100 °C), where the prediction accuracy does not conform to this rule when the strain rate is 0.1 s^−1^ and the temperature is 1050 °C. An ideal constitutive model should have the same prediction accuracy at different temperatures and strain rates. Therefore, the standard Arrhenius constitutive equation needs to be improved when the prediction accuracy of the rheological properties of α+β titanium alloys is required to be high.

### 3.2. Hensel–Spittel Model

The Hensel–Spittel (HS) constitutive model, which has been used in famous commercial software (Forge NxT), is frequently used in hot forming simulation. Its form is simple and its parameters can be obtained easily [30,31]. The general form of the HS model is exhibited in Equation (4).
(4)$$\sigma =\mathrm{Aexp}\left(m_{1}T\right)\varepsilon^{m_{2}}{\dot{\varepsilon }}^{m_{3}}\exp\left(\frac{m_{4}}{\varepsilon }\right){\left(1+\varepsilon \right)}^{m_{5}T}\exp\left(m_{6}\varepsilon \right){\dot{\varepsilon }}^{m_{7}T}T^{m_{8}}$$
where ε, σ, $\dot{\varepsilon }$, T are strain, flow stress, strain rate and temperature, A and $m_{1}$–$m_{8}$ are material constants. Equation (5) can be obtained by taking the natural logarithm of Equation (4).
(5)$$\ln\sigma =\ln A+m_{1}T+m_{2}\ln\varepsilon +m_{3}\ln\dot{\varepsilon }+m_{4}/\varepsilon +m_{5}T\ln\left(1+\varepsilon \right)+m_{6}\varepsilon \;+m_{7}T\ln\dot{\varepsilon }+m_{8}\ln T$$

There is a linear relationship between lnσ, T, lnε, $\ln\dot{\varepsilon }$, 1/ε, Tln(1+ε), ε, $T\ln\dot{\varepsilon }$, and lnT. Then, the solving problem of the material constants in Equation (5) is a typical multiple linear regression problem. Similarly, the strain has been divided into 10 parts between 0.04 and 0.9. Then, the flow stress at each strain level can be obtained by interpolating the original compressed data (shown in Figure 1). The stress, strain, temperature and strain rate corresponding to all of the strain levels should approximately satisfy Equation (5) (not absolutely satisfied due to regression error). All strains are brought into Equation (5) and a random error factor is introduced to obtain linear simultaneous Equation (6).
(6)$$\left[\begin{array}{c} \ln\sigma_{1}\\ \ln\sigma_{2}\\ \begin{array}{c} \ln\sigma_{3}\\ \vdots \\ \ln\sigma_{n} \end{array} \end{array}\right]=[\begin{array}{c} 1\\ 1\\ \begin{array}{c} 1\\ \vdots \\ 1 \end{array} \end{array}\;\begin{array}{c} T_{1}\\ T_{2}\\ \begin{array}{c} T_{3}\\ \vdots \\ T_{n} \end{array} \end{array}\;\begin{array}{c} \ln\varepsilon_{1}\\ \ln\varepsilon_{2}\\ \begin{array}{c} \ln\varepsilon_{3}\\ \vdots \\ \ln\varepsilon_{n} \end{array} \end{array}\;\begin{array}{c} \ln{\dot{\varepsilon }}_{1}\\ \ln{\dot{\varepsilon }}_{2}\\ \begin{array}{c} \ln{\dot{\varepsilon }}_{3}\\ \vdots \\ \ln{\dot{\varepsilon }}_{n} \end{array} \end{array}\;\begin{array}{c} 1/\varepsilon_{1}\\ 1/\varepsilon_{2}\\ \begin{array}{c} 1/\varepsilon_{3}\\ \vdots \\ 1/\varepsilon_{n} \end{array} \end{array}\;\begin{array}{c} T_{1}\ln\left(1+\varepsilon_{1}\right)\\ T_{2}\ln\left(1+\varepsilon_{2}\right)\\ \begin{array}{c} T_{3}\ln\left(1+\varepsilon_{3}\right)\\ \vdots \\ T_{n}\ln\left(1+\varepsilon_{n}\right) \end{array} \end{array}\begin{array}{c} \varepsilon_{1}\\ \varepsilon_{2}\\ \begin{array}{c} \varepsilon_{3}\\ \vdots \\ \varepsilon_{n} \end{array} \end{array}\;\begin{array}{c} T_{1}\ln{\dot{\varepsilon }}_{1}\\ T_{2}\ln{\dot{\varepsilon }}_{2}\\ \begin{array}{c} T_{3}\ln{\dot{\varepsilon }}_{3}\\ \vdots \\ T_{n}\ln{\dot{\varepsilon }}_{n} \end{array} \end{array}\;\begin{array}{c} \ln T_{1}\\ \ln T_{2}\\ \begin{array}{c} \ln T_{3}\\ \vdots \\ \ln T_{n} \end{array} \end{array}]\gamma +\mu$$
where *n* = 5 × 5 × 10 (5 temperatures, 5 strain rates and 10 strain levels), $\gamma ={\left[\ln A\;m_{1}\;m_{2}\;m_{3}\;m_{4}\;m_{5}\;m_{6}\;m_{7}\;m_{8}\right]}^{\prime }$, μ is the error vector with size of *n* × 1, which follows the normal distribution with the mean value of zero. Equation (6) is an overdetermined equation in which the number of variables is fewer than the number of equations. The least square method is a classical method used to solve this overdetermined equation. The γ can be solved by using function of “γ = regress(Y, X)” in the software of MATLAB. The multiple linear regression results of the material parameters are listed in Table 4, and the comparisons of the HS model prediction flow stress and experimental flow stress are shown in Figure 5.

As shown in Figure 5, the prediction accuracy of the HS constitutive equation is low in the single-phase region (900, 950 and 100 °C) and high in the two-phase region (1050 and 1100 °C). This rule is the same as for the Arrhenius model. The prediction accuracy of the HS model is lower than that of the Arrhenius model, but the HS model has only 9 material parameters and the Arrhenius model has 24 material parameters. Generally, the more parameters of the model, the higher the prediction accuracy. So far, it is difficult to improve the prediction accuracy of either model. Based on this, a new constitutive model is urgently needed.

## 4. New Constitutive Model

### 4.1. Mathematical Principles

An idea can be drawn from the Arrhenius model; that is, when the strain is fixed, the core of building the constitutive model of materials is to build the functional relationship between stress, strain rate and temperature. The Arrhenius model is an implicit equation, which is complicated in practical application. In addition, the prediction accuracy of this model is also low for the Ti6242s alloy. In fact, the relationship between stress, strain rate and temperature can be defined as $\sigma =g\left(\dot{\varepsilon },T\right)$. However, when considering the large nonlinear relationship among stress, strain rate and temperature, this function is not used. The effects of strain rate and temperature on stress were analyzed before proposing a new constitutive model. The data for low (ε=0.04), medium (ε=0.23) and high (ε=0.90) strain levels in Table 2 were selected for analysis. In order to study the relationship between logarithmic stress, logarithmic strain rate and temperature, the partial derivatives of logarithmic stress with respect to temperature and logarithmic strain rate can be calculated by discrete formula, that is, the difference quotient replacing the derivative. The discrete formulas of derivatives include forward difference, backward difference and central difference. In order to improve the accuracy, forward and backward difference are used at the boundary, and central difference is used in the middle region. Then, the nth-order partial derivatives of logarithmic stress with respect to the temperature and logarithmic strain rate for the experimental flow stress curves can be calculated by Equation (7).
(7)$$\left\{ \begin{array}{c} {\frac{\partial^{n}\ln\sigma }{\partial T^{n}}|}_{i,j}=\frac{{\frac{\partial^{n-1}\ln\sigma }{\partial T^{n-1}}|}_{i,j+1}-{\;\frac{\partial^{n-1}\ln\sigma }{\partial T^{n-1}}|}_{i,j-1}}{T_{i,j+1}-T_{i,j-1}}\\ \begin{array}{c} {\frac{\partial^{n}\ln\sigma }{\partial {\left(\ln\dot{\varepsilon }\right)}^{n}}|}_{i,j}=\frac{{\frac{\partial^{n-1}\ln\sigma }{\partial {\left(\ln\dot{\varepsilon }\right)}^{n-1}}|}_{i+1,j}-{\;\frac{\partial^{n-1}\ln\sigma }{\partial {\left(\ln\dot{\varepsilon }\right)}^{n-1}}|}_{i-1,j}}{\ln{\dot{\varepsilon }}_{i+1,j}-\ln{\dot{\varepsilon }}_{i-1,j}} \end{array} \end{array}\right.$$
where n is the order of the partial derivative, i and j are indices of the strain rate and the temperature in the stress matrix, respectively (Table 3). For example, when i = 1 and j = 1, $\ln\sigma_{i,j}$ represents the logarithmic stress with a strain rate of 0.001 s^−1^ and a temperature of 900 °C. The maximum values of i and j, which are both five in this study, are the number of strain rates and temperatures corresponding to the experiment, respectively. Equation (7) can be replaced by Equation (8) when i = 1 and j = 1.


(8)
$$\left\{ \begin{array}{c} {\frac{\partial^{n}\ln\sigma }{\partial T^{n}}|}_{i,j}=\frac{{\frac{\partial^{n-1}\ln\sigma }{\partial T^{n-1}}|}_{i,j+1}-{\;\frac{\partial^{n-1}\ln\sigma }{\partial T^{n-1}}|}_{i,j}}{T_{i,j+1}-T_{i,j}}\\ {\frac{\partial^{n}\ln\sigma }{\partial {\left(\ln\dot{\varepsilon }\right)}^{n}}|}_{i,j}=\frac{{\frac{\partial^{n-1}\ln\sigma }{\partial {\left(\ln\dot{\varepsilon }\right)}^{n-1}}|}_{i+1,j}-\;{\frac{\partial^{n-1}\ln\sigma }{\partial {\left(\ln\dot{\varepsilon }\right)}^{n-1}}|}_{i,j}}{\ln{\dot{\varepsilon }}_{i+1,j}-\ln{\dot{\varepsilon }}_{i,j}} \end{array}\right.$$


Equation (7) can be replaced by Equation (9) when i = 5 and j = 5.


(9)
$$\left\{ \begin{array}{c} {\frac{\partial^{n}\ln\sigma }{\partial T^{n}}|}_{i,j}=\frac{{\frac{\partial^{n-1}\ln\sigma }{\partial T^{n-1}}|}_{i,j}-\;{\frac{\partial^{n-1}\ln\sigma }{\partial T^{n-1}}|}_{i,j-1}}{T_{i,j}-T_{i,j-1}}\\ {\frac{\partial^{n}\ln\sigma }{\partial {\left(\ln\dot{\varepsilon }\right)}^{n}}|}_{i,j}=\frac{{\frac{\partial^{n-1}\ln\sigma }{\partial {\left(\ln\dot{\varepsilon }\right)}^{n-1}}|}_{i,j}-\;{\frac{\partial^{n-1}\ln\sigma }{\partial {\left(\ln\dot{\varepsilon }\right)}^{n-1}}|}_{i-1,j}}{\ln{\dot{\varepsilon }}_{i,j}-\ln{\dot{\varepsilon }}_{i-1,j}} \end{array}\right.$$


Equation (8) can be replaced by Equation (10) when n = 1.
(10)$$\{\begin{array}{c} {\frac{\partial \ln\sigma }{\partial T}|}_{i,j}=\frac{\ln\sigma_{i,j+1}-\ln\sigma_{i,j-1}}{T_{i,j+1}-T_{i,j-1}}\\ {\frac{\partial \ln\sigma }{\partial \ln\dot{\varepsilon }}|}_{i,j}=\frac{\ln\sigma_{i+1,j}-\ln\sigma_{i-1,j}}{\ln{\dot{\varepsilon }}_{i+1,j}-\ln{\dot{\varepsilon }}_{i-1,j}} \end{array}$$

In fact, the greater the number of the temperatures and the strain rates in the experiment, the higher the calculation accuracy of Equations (7)–(10) is. As shown in Table 5 and Table 6, the first, second and third partial derivatives of logarithmic stress with respect to the temperature and logarithmic strain rate at low (ε=0.04), medium (ε=0.23) and high (ε=0.90) strain levels were calculated, using Equations (7)–(10). The redder the color in the figure, the smaller the value represented.

There are several phenomena that can be found from Table 5: (1) with the increase in temperature, the logarithmic stress has a monotonic decreasing characteristic for all strain levels (Table 5 a, e and i). (2) The difference of the first (∂lnσ/∂T) partial derivative of logarithmic stress with respect to the temperature is small for a certain strain level (Table 5 b, f, and j). (3) The second ($\partial^{2}\ln\sigma /\partial T^{2}$) and third ($\partial^{3}\ln\sigma /\partial T^{3}$) partial derivatives of logarithmic stress with respect to the temperature is close to zero for all strain levels (Table 5 c, g, k, d, h and l). According to calculus theory and the above phenomena, a basic conclusion can be derived; that is, the linear model can construct the relationship between logarithmic stress and temperature with high accuracy.

Similarly, there are several phenomena that can be found from Table 6 as follows: (1) with the increase in logarithmic strain rate, the logarithmic stress has a monotonic decreasing characteristic for all strain levels (see a, e and i in Table 6). (2) The difference in the first ($\partial \ln\sigma /\partial \ln\dot{\varepsilon }$) partial derivative of logarithmic stress with respect to the logarithmic strain rate is relatively large for all strain levels (Table 6 b, f and j), and with the increase in temperature and logarithmic strain rate, the first ($\partial \ln\sigma /\partial \ln\dot{\varepsilon }$) partial derivative is increased. (3) The second ($\partial^{2}\ln\sigma /\partial {\left(\ln\dot{\varepsilon }\right)}^{2}$) partial derivative of logarithmic stress with respect to the logarithmic strain rate is relatively large for all strain levels (Table 6 c, g and k), and the third ($\partial^{3}\ln\sigma /\partial {\left(\ln\dot{\varepsilon }\right)}^{3}$) partial derivative of logarithmic stress with respect to the logarithmic strain rate is very small (Table 6 d, h and l). According to calculus theory and the above phenomena, a basic conclusion can be derived; that is, the quadratic model can construct the relationship between logarithmic stress and logarithmic strain rate with high accuracy.

The function of logarithmic stress, logarithmic strain rate and temperature has been defined as $\ln\sigma =f\left(\ln\dot{\varepsilon },T\right)$. According to the binary Taylor expansion formula, $f\left(\ln\dot{\varepsilon },T\right)$ is expanded at $\left(\ln\dot{\varepsilon }=0,T=0\right)$ to obtain Equation (11).
(11)$$\ln\sigma =\overset{m}{\underset{i=0}{\sum }}\left[\frac{1}{m!}\overset{m}{\underset{n=0}{\sum }}\left(C_{m}^{n}{\left(\ln\dot{\varepsilon }\right)}^{n}T^{m-n}\left({\frac{\partial^{m}f}{\partial {\left(\ln\dot{\varepsilon }\right)}^{n}\partial T^{m-n}}|}_{\left(0,0\right)}\right)\right)\right]$$
where *m* is the number of terms, $C_{m}^{n}$ is a combination operator and $\partial^{m}$ is m-order partial derivative operator. Equation (11) shows that function f can be approximated by polynomials. The more complex the nonlinear relationship is, the more items are needed to build the same precision model. It is a linear model when *m* = 1, it is a quadratic model when *m* = 2, and so on. The greater the *m* is, the higher the accuracy is. It can be predicted that if the rheological data of the materials are relatively complex (such as including single-phase and two-phase microstructures, especially for α+β titanium alloy), m should be appropriately increased. According to Table 5 and Table 6, $\partial^{3}\ln\sigma /\partial T^{3}$ and $\partial^{3}\ln\sigma /\partial \ln{\dot{\varepsilon }}^{3}$ of Ti6242s alloy are both close to zero. It can be asserted that the quadratic model can be used to build a high-precision constitutive model $f\left(\ln\dot{\varepsilon },T\right)$ of this material.

### 4.2. Linear Model

First, the accuracy of the first-order model (linear model) was analyzed. When *m* = 1, the constitutive Equation (11) is simplified to Equation (12).
(12)$$\ln\sigma =k_{0}+k_{1}T+k_{2}\ln\dot{\varepsilon }$$
where $k_{0}$, $k_{1}$ and $k_{2}$ are material parameters, which can be obtained by using multiple linear regression based on Table 3. The linear regression expression of material parameters corresponding to each strain is shown in Equation (13).
(13)$$\left[\begin{array}{c} \ln\sigma_{1}\\ \ln\sigma_{2}\\ \begin{array}{c} \ln\sigma_{3}\\ \vdots \\ \ln\sigma_{w} \end{array} \end{array}\right]=\left[\begin{array}{c} 1\\ 1\\ \begin{array}{c} 1\\ \vdots \\ 1 \end{array} \end{array}\;\begin{array}{c} T_{1}\\ T_{2}\\ \begin{array}{c} T_{3}\\ \vdots \\ T_{w} \end{array} \end{array}\;\begin{array}{c} \ln{\dot{\varepsilon }}_{1}\\ \ln{\dot{\varepsilon }}_{2}\\ \begin{array}{c} \ln{\dot{\varepsilon }}_{3}\\ \vdots \\ \ln{\dot{\varepsilon }}_{w} \end{array} \end{array}\;\right]\left[\begin{array}{c} k_{0}\\ k_{1}\\ k_{2} \end{array}\right]+\mu_{1}$$
where w is the combined number of temperatures and strain rates when the strain is fixed (*w* = 5 × 5 in this study), and $\mu_{1}$ is an error variable and follows the normal distribution with the mean value of zero. Ten groups of material parameters can be obtained by linear regression for each strain datum, which are shown as points in Figure 6.

The expression of each material parameter and strain can be obtained by fitting the data points of linear regression with a fifth-order polynomial. Then, the expression of each material is shown in Equation (14).
(14)$$\{\begin{array}{c} k_{0}=10.7957+26.6315\varepsilon -124.7369\varepsilon^{2}+261.1311\varepsilon^{3}-285.3326\varepsilon^{4}+97.1624\varepsilon^{5}\\ k_{1}=-0.0060-0.0249\varepsilon +0.1154\varepsilon^{2}-0.2428\varepsilon^{3}+0.2519\varepsilon^{4}-0.0916\varepsilon^{5}\\ k_{2}=0.1634+0.1248\varepsilon -0.2586\varepsilon^{2}+0.1393\varepsilon^{3}+0.2671\varepsilon^{4}-0.2520\varepsilon^{5} \end{array}$$

The linear constitutive equation of the titanium alloy can be obtained by bringing Equation (14) into Equation (12). The comparisons of prediction flow stress and experimental flow stress for the linear model are shown in Figure 7.

This model has a similar phenomenon to the Arrhenius model and HS model, that is, the prediction accuracy is poor in the low temperature region (900, 950 and 1000 °C) and higher in the high temperature region (1050 and 1100 °C). The model has the following advantages: fewer material parameters, a simple form and easy solution of material parameters (multiple linear regression only). In order to further improve the accuracy, a high-order constitutive model was proposed.

### 4.3. Quadratic Model

The linear model has low prediction accuracy for low temperature regions. Therefore, a quadratic model was proposed, in which the m is equal to two. The equation of the quadratic model is as follows:(15)$$\ln\sigma =a_{0}+a_{1}\ln\dot{\varepsilon }+a_{2}T+a_{3}{\left(\ln\dot{\varepsilon }\right)}^{2}+a_{4}T^{2}+a_{5}T\ln\dot{\varepsilon }$$
where $a_{0}\sim a_{5}$ are material parameters, which can be obtained by multiple linear regression, based on Table 3. The linear regression expression of the material parameters corresponding to each strain is shown in Equation (16).
(16)$$\left[\begin{array}{c} \ln\sigma_{1}\\ \ln\sigma_{2}\\ \begin{array}{c} \ln\sigma_{3}\\ \vdots \\ \ln\sigma_{w} \end{array} \end{array}\right]=\left[\begin{array}{c} 1\\ 1\\ \begin{array}{c} 1\\ \vdots \\ 1 \end{array} \end{array}\;\begin{array}{c} \ln{\dot{\varepsilon }}_{1}\\ \ln{\dot{\varepsilon }}_{2}\\ \begin{array}{c} \ln{\dot{\varepsilon }}_{3}\\ \vdots \\ \ln{\dot{\varepsilon }}_{w} \end{array} \end{array}\;\begin{array}{c} T_{1}\\ T_{2}\\ \begin{array}{c} T_{3}\\ \vdots \\ T_{w} \end{array} \end{array}\;\begin{array}{c} {\left(\ln{\dot{\varepsilon }}_{1}\right)}^{2}\\ {\left(\ln{\dot{\varepsilon }}_{2}\right)}^{2}\\ \begin{array}{c} {\left(\ln{\dot{\varepsilon }}_{3}\right)}^{2}\\ \vdots \\ {\left(\ln{\dot{\varepsilon }}_{w}\right)}^{2} \end{array} \end{array}\;\begin{array}{c} T_{1}{}^{2}\\ T_{2}{}^{2}\\ \begin{array}{c} T_{3}{}^{2}\\ \vdots \\ T_{w}{}^{2} \end{array} \end{array}\;\begin{array}{c} T_{1}\ln{\dot{\varepsilon }}_{1}\\ T_{2}\ln{\dot{\varepsilon }}_{2}\\ \begin{array}{c} T_{3}\ln{\dot{\varepsilon }}_{3}\\ \vdots \\ T_{w}\ln{\dot{\varepsilon }}_{w} \end{array} \end{array}\;\right]{\left[a_{0}\;a_{1}\;a_{2}\;a_{3}\;a_{4}\;a_{5}\right]}^{'}+\mu_{2}$$
where w is the combined number of temperatures and strain rates when the strain is fixed (*w* = 5 × 5 in this study), and $\mu_{2}$ is an error variable and follows the normal distribution with the mean value of zero. Ten groups of material parameters can be obtained by linear regression for each strain data, which are shown as points in Figure 8.

The expression of each material parameter and strain can be obtained by fitting the data points of linear regression with a fifth-order polynomial. Then, the expression of each material is shown in Equation (17).


(17)
$$\{\begin{array}{l} a_{0}=1.58977+243.78788\varepsilon -1176.81606\varepsilon^{2}+2530.31159\varepsilon^{3}-2516.33047\varepsilon^{4}+939.51775\varepsilon^{5}\\ a_{1}=-0.217555-0.222702\varepsilon +2.174965\varepsilon^{2}-2.618603\varepsilon^{3}-1.456050\varepsilon^{4}+2.177008\varepsilon^{5}\\ a_{2}=0.011670-0.461668\varepsilon +2.234179\varepsilon^{2}-4.807239\varepsilon^{3}+4.773571\varepsilon^{4}-1.777977\varepsilon^{5}\\ a_{3}=-0.004285-0.022862\varepsilon -0.075039\varepsilon^{2}-0.146988\varepsilon^{3}+0.141039\varepsilon^{4}-0.053276\varepsilon^{5}\\ a_{4}=0.000008+0.000218\varepsilon -0.001061\varepsilon^{2}+0.002284\varepsilon^{3}-0.002263\varepsilon^{4}+0.0008400\varepsilon^{5}\\ a_{5}=0.000361+0.000242\varepsilon -0.002088\varepsilon^{2}+0.002081\varepsilon^{3}+0.002373\varepsilon^{4}-0.002674\varepsilon^{5} \end{array}$$


Similarly, we inserted (17) into (15) to obtain the constitutive equation of the quadratic model, and compared the predicted values of the model with the experimental values to give Figure 9.

Compared with the linear model (see Figure 7 and Figure 9), the prediction accuracy of the quadratic model was improved significantly. In addition, the prediction accuracy of the quadratic model is significantly higher than that of the HS model and the Arrhenius model. Contrary to the previous models, the prediction accuracy of the model is high in both the single-phase region (900, 950 and 1000 °C) and the two-phase region (1050 and 1100 °C).

## 5. Predictability Analysis

### 5.1. Prediction Accuracy

#### 5.1.1. Overall Accuracy

In order to analyze the overall prediction accuracy, the accuracy of the classical models and the new models was quantified in terms of standard statistical parameters such as correlation coefficient (R), root mean square error (RMSE), sum of squares for error (SSE), and sum of absolute error (SAE). The formulas of these parameters are shown in Equations (18)–(21).
(18)$$R=\frac{\sum_{i=1}^{N}\left({\hat{\sigma }}_{i}-\bar{\hat{\sigma }}\right)\left(\sigma_{i}-\bar{\sigma }\right)}{\sqrt{\sum_{i=1}^{N}{\left({\hat{\sigma }}_{i}-\bar{\hat{\sigma }}\right)}^{2}\sum_{i=1}^{N}{\left(\sigma_{i}-\bar{\sigma }\right)}^{2}}}$$
(19)$$RMSE=\sqrt{\frac{\sum_{i=1}^{N}{\left({\hat{\sigma }}_{i}-\sigma_{i}\right)}^{2}}{N}}$$
(20)$$SSE=\overset{N}{\underset{i=1}{\sum }}{\left({\hat{\sigma }}_{i}-\sigma_{i}\right)}^{2}$$
(21)$$SAE=\overset{N}{\underset{i=1}{\sum }}\left|{\hat{\sigma }}_{i}-\sigma_{i}\right|$$
where ${\hat{\sigma }}_{i}$ is the ith prediction value of the flow stress, $\sigma_{i}$ is the ith experiment value of flow stress, $\bar{\hat{\sigma }}$ is the mean value of the prediction flow stress, $\bar{\sigma }$ is the mean value of the experiment flow stress, and N is the number of comparison points (*N* = 5 × 5 × 10). The quantitative comparison of the prediction accuracy of the different models is listed in Table 7.

As can be seen from Table 7, the order of prediction accuracy from high to low is the quadratic model, Arrhenius model, HS model and linear model. The SSE of the quadratic model (1.0E3) is one order of magnitude smaller than the other models (1.0E5). The RMSE of the quadratic model is 1/2 of the Arrhenius model, 1/3 of the HS model and 1/4 of the linear model. The correlation coefficients of the quadratic model are significantly higher than the other models. The sum of absolute error of the quadratic model is 884.17 MPa, which is much less than the other models (1422.53 MPa for the Arrhenius model, 2097.60 MPa for the HS model and 2827.67 MPa for the linear model). Therefore, the overall accuracy of the quadratic model is significantly higher than the other models.

As can be seen from Figure 10, the data points of the quadratic model are distributed on both sides of the centerline and more concentrated than the other models. In addition, the square of the correlation coefficient between experimental stress and predicted stress in the quadratic model, Arrhenius model and Hensel–Spittel model are 0.9961, 0.988 and 0.9751, respectively. This also shows that the prediction accuracy of the quadratic model is higher than that of the other models.

As shown in Figure 11, the “•” symbol represents the predictive value of the Arrhenius model, the “□” symbol represents the predictive value of the linear model, the “×” represents the predictive value of the HS model, the “+” represents the predictive value of the quadratic model, and the solid line represents the experimental curve. Almost all the prediction points of the quadratic model are closer to the experimental curve than the other models. The precision of the quadratic model is significantly higher than that of the other models.

#### 5.1.2. Local Accuracy

In order to analyze the prediction accuracy under different temperatures and strain rates, the correlation coefficient (R) and sum of absolute error (SAE) were calculated. The distribution of the correlation coefficient between predicted flow stress and experimental flow stress on temperature and strain rate is shown in Figure 12.

As shown in Figure 12, the difference from high to low in the correlation coefficient with temperature gives the ranking order as the HS model, Arrhenius model, linear model and quadratic model. The difference between the maximum value and the minimum value of the correlation coefficient for the quadratic model was 0.073, which was significantly less than that of the other models (0.2486 for HS model, 0.1228 for linear model and 0.1199 for Arrhenius model). The rules in Figure 12 show that the prediction accuracy of Arrhenius model and Hensel–Spittel model in the two phase region (α+β) and single phase (α) is significantly different. The prediction accuracy of the quadratic model remains high in both the single phase region and the two phase region, and the prediction accuracy of the quadratic model is significantly higher than the other models. The prediction accuracy of the linear model in the single phase region and the two phase region is also similar to the quadratic model, but the prediction accuracy is lower. In addition, the distribution of the correlation coefficient on the strain rate for all models also has the above rules, that is, the distribution of the correlation coefficient on strain rate for the Hensel–Spittel and Arrhenius models is very different. The correlation coefficient of the quadratic model at different strain rates and temperatures is large, and the difference is small.

The distribution of the sum of absolute error between predicted flow stress and experimental flow stress on temperature and strain rate is shown in Figure 13.

As shown in Figure 13, the maximum absolute error sums of the Arrhenius model, HS model, linear model and quadratic model are 237.4, 274.3, 727.2, and 95.53 MPa, respectively. This also shows that the accuracy of the quadratic model is significantly higher than that of the other models. In addition, the maximum and minimum values of the sum of absolute error for the Arrhenius model, HS model and linear model differ greatly, which indicates that the prediction accuracy of these three models for specific strain rate and temperature is poor. The distribution of the absolute error sum of the quadratic model is relatively uniform, which shows that the quadratic model has good prediction ability for all temperatures and strain rates.

### 5.2. Predictability of Model

Generally, linear and non-linear multiple regression can be used to obtain the global optimal material parameters for classical models and the linear model. However, the classical models and the linear model still produce large errors. Therefore, the reason for the large error could be that the classical models and the linear model have insufficient predictability. In this section, mathematical means are used to study the reasons for the insufficient precision of the classical models and the linear model.

#### 5.2.1. Analytic Formula of Partial Differential

It can be seen from Table 5 and Table 6 that the third derivative of the logarithmic stress with respect to the logarithmic strain rate of the material is close to zero, and the second derivative of the logarithmic stress with respect to the temperature is close to zero. Therefore, the analytical partial derivatives of all models are derived first. The first-, second- and third-order partial derivatives of the logarithmic stress with respect to the logarithmic strain rate and temperature for all constitutive models can be solved by Equation (22).
(22)$$\{\begin{array}{l} \frac{\partial \ln\sigma }{\partial \ln\dot{\varepsilon }}=\frac{\dot{\varepsilon }\partial \sigma }{\sigma \partial \dot{\varepsilon }}\\ \frac{\partial \ln\sigma }{\partial T}=\frac{\partial \sigma }{\sigma \partial T}\\ \frac{\partial^{2}\ln\sigma }{\partial {\left(\ln\dot{\varepsilon }\right)}^{2}}=\frac{\dot{\varepsilon }\partial \left(\frac{\partial \ln\sigma }{\partial \ln\dot{\varepsilon }}\right)}{\sigma \partial \dot{\varepsilon }}\\ \frac{\partial^{2}\ln\sigma }{\partial T^{2}}=\frac{\partial \left(\frac{\partial \ln\sigma }{\partial T}\right)}{\sigma \partial T}\;\;\\ \frac{\partial^{3}\ln\sigma }{\partial {\left(\ln\dot{\varepsilon }\right)}^{3}}=\frac{\dot{\varepsilon }\partial \left(\frac{\partial^{2}\ln\sigma }{\partial {\left(\ln\dot{\varepsilon }\right)}^{2}}\right)}{\sigma \partial \dot{\varepsilon }}\\ \frac{\partial^{3}\ln\sigma }{\partial T^{3}}=\frac{\partial \left(\frac{\partial^{2}\ln\sigma }{\partial T^{2}}\right)}{\sigma \partial T}\;\;\; \end{array}$$
(23)$$\sigma =\{\begin{array}{c} \frac{1}{\alpha }\ln\left({\left(\frac{1}{A}\dot{\varepsilon }\exp\left(\frac{Q}{RT}\right)\right)}^{\frac{1}{n}}+\sqrt{{\left(\frac{1}{A}\dot{\varepsilon }\exp\left(\frac{Q}{RT}\right)\right)}^{\frac{2}{n}}+1}\right),\;\mathrm{for}\;\mathrm{Arrhenius}\;\mathrm{Model}\\ \mathrm{Aexp}\left(m_{1}T\right)\varepsilon^{m_{2}}{\dot{\varepsilon }}^{m_{3}}\exp\left(\frac{m_{4}}{\varepsilon }\right){\left(1+\varepsilon \right)}^{m_{5}T}\exp\left(m_{6}\varepsilon \right){\dot{\varepsilon }}^{m_{7}T}T^{m_{8}},\;\mathrm{for}\;\mathrm{HS}\;\mathrm{Model}\;\\ \exp\left(k_{0}+k_{1}T+k_{2}\ln\dot{\varepsilon }\right),\;\mathrm{for}\;\mathrm{Linear}\;\mathrm{Model}\;\\ \exp\left(a_{0}+a_{1}\ln\dot{\varepsilon }+a_{2}T+a_{3}{\left(\ln\dot{\varepsilon }\right)}^{2}+a_{4}T^{2}+a_{5}T\ln\dot{\varepsilon }\right),\;\mathrm{for}\;\mathrm{Quadratic}\;\mathrm{Model} \end{array}$$
where A, α, n and Q are quintic polynomials of ε shown in Equation (3), lnA and $m_{1}$–$m_{8}$ are constants shown in Table 4. $k_{0}$–$k_{1}$ are quintic polynomials of ε shown in Equation (14) and $a_{0}$–$a_{5}$ are quintic polynomials of ε shown in Equation (17). When the strain value is fixed, these parameters are constants.

#### 5.2.2. Results of Partial Differential

The first, second and third partial derivatives of logarithmic stress with respect to temperature for experimental data and different models are listed in Table 8 when strain is equal to 0.04 by using Equations (22) and (23). The redder the color in the figure, the smaller the value represented.

There are several rules that can be drawn from Table 8: (1) the second and third partial derivatives of all models are close to zero, which are in good agreement with the experimental data (Table 8 c, g, k, o, s, d, h, l, p and t). (2) The first partial derivatives for the Arrhenius model and the HS model have great regularity with temperature. The partial differential decreases with the increase in temperature for the Arrhenius model and the law is the opposite for the HS model (Table 8 f and j). (3) The first partial derivative for the linear model is not related to temperature or strain rate (Table 8 n). (4) The first partial derivative for the quadratic model is less dependent on temperature, but strongly dependent on strain rate (Table 8 r). (5) The first partial derivative for the experimental data is less dependent on temperature, but slightly dependent strain rate (Table 8 b). The above phenomena show that the experimental data can be approximated well only by one order accuracy, and all models meet the first-order accuracy. However, the quadratic model has second-order accuracy between logarithmic stress and temperature. Considering that all models have first-order accuracy for temperature, another error source is the strain rate factor.

The first, second and third partial derivatives of logarithmic stress with respect to logarithmic strain rate for experimental data and different models are listed in Table 9, when strain is equal to 0.04, by using Equations (22) and (23). The redder the color in the figure, the smaller the value represented. It can be seen from Table 9 that the second-order partial derivatives of logarithmic stress with respect to logarithmic strain rate for the Arrhenius model, the HS model and the linear model are close to zero. It is worth noting that the third partial differential of logarithmic stress with respect to logarithmic strain rate for the quadratic model is close to zero. This phenomenon is determined by the equation of the constitutive model and is independent of the regression error, because the partial derivatives are calculated by analytical formulas. However, the second derivative of logarithmic stress with respect to logarithmic strain rate for the experimental data is not close to zero. This proves that the Arrhenius model, the HS model and the linear model cannot make a second-order approximation to the experimental data at logarithmic strain rate, while the quadratic model can.

By comparing the first, second and third partial derivatives of all models, the following conclusions can be drawn: (1) the predictabilities of the classical models are only the first-order accuracy in the logarithmic stress, logarithmic strain rate and temperature space. (2) In essence, the predictabilities of the Arrhenius model, the HS model and the linear model are in the same order of magnitude, because they are linear approximations of the logarithmic stress, logarithmic strain rate and temperature space. (3) The predictability of the quadratic model is one level higher than that of the Arrhenius model, the HS model and the linear model, because it is a quadratic approximation. (4) The traditional modification methods for the Arrhenius and HS models cannot significantly improve the predictability of the models, because these model modifications are carried out within the linear level in the logarithmic stress, logarithmic strain rate and temperature space. (5) Due to the existence of the cross term of temperature and logarithmic strain rate ($T\ln\dot{\varepsilon }$) and the quadratic term of temperature ($T^{2}$), the quadratic model can largely eliminate the predictability difference between the classical model and the linear model at different temperatures.

## 6. Conclusions

In this paper, hot compression tests under different strain rates and temperatures were carried out. A new constitutive model based on the Taylor series and partial derivatives of the experimental data is proposed without significantly increasing the material parameters. The predictabilities of the new model and the classical models (Arrhenius and HS models) were analyzed. The reason for the lack of predictability in the classical models is proved mathematically. The main conclusions are as follows:

(1) The prediction accuracies of the Arrhenius model and HS model are significantly different in different temperatures and with different strain rates. The prediction accuracies of the Arrhenius model and the HS model are low in the single-phase region (900, 950 and 100 °C) and high in the two-phase region (1050 and 1100 °C). The prediction accuracy of the quadratic model in all temperatures and with all strain rates has no significant difference, and is higher than the other models. The order of prediction accuracy from high to low is the quadratic model, the Arrhenius model, the HS model and the linear model. The SSE of the quadratic model (1.0 × 10^3^) is one order of magnitude smaller than the other models (1.0 × 10^5^). The RMSE of the quadratic model is 1/2 of the Arrhenius model, 1/3 of the HS model and 1/4 of the linear model. The correlation coefficients of the quadratic model are significantly high than the other models. The sum of absolute error of the quadratic model is 884.17 MPa which is much less than the other models (1422.53 MPa for the Arrhenius model, 2097.60 MPa for the HS model and 2827.67 MPa for the linear model). The square of the correlation coefficient between experimental stress and predicted stress in the quadratic model, Arrhenius model and Hensel–Spittel model is 0.9961, 0.988 and 0.9751, respectively. This also shows that the prediction accuracy of the quadratic model is higher than that of the other models.

(2) The second and third partial derivatives of logarithmic stress with respect to temperature for the experimental data is close to zero. The third partial derivatives of logarithmic stress with respect to logarithmic strain rate for the experimental data is close to zero. This phenomena indicates that second-order approximation is required between logarithmic stress and logarithmic strain rate, and first-order approximation is required between logarithmic stress and temperature to establish a high-precision model. The Arrhenius model, HS model and linear model meet the first-order approximation requirements between logarithmic stress and temperature, but do not meet the second-order approximation requirements between logarithmic stress and logarithmic strain rate. The quadratic model meets these two requirements and thus has higher prediction accuracy than the other models.

(3) The order of prediction accuracy for the four models from high to low is the quadratic model, Arrhenius model, linear model and HS model. The material parameters of the quadratic model can be solved only by multiple linear regression, while the Arrhenius model needs to use multiple nonlinear regression. Compared with the Arrhenius model, the prediction accuracy of the quadratic model can be improved significantly by adding only a few parameters, and the parameter solution is simpler. The prediction accuracy of the new model can be further improved by adding the number of items.

## Figures and Tables

**Figure 1 materials-16-02928-f001:**
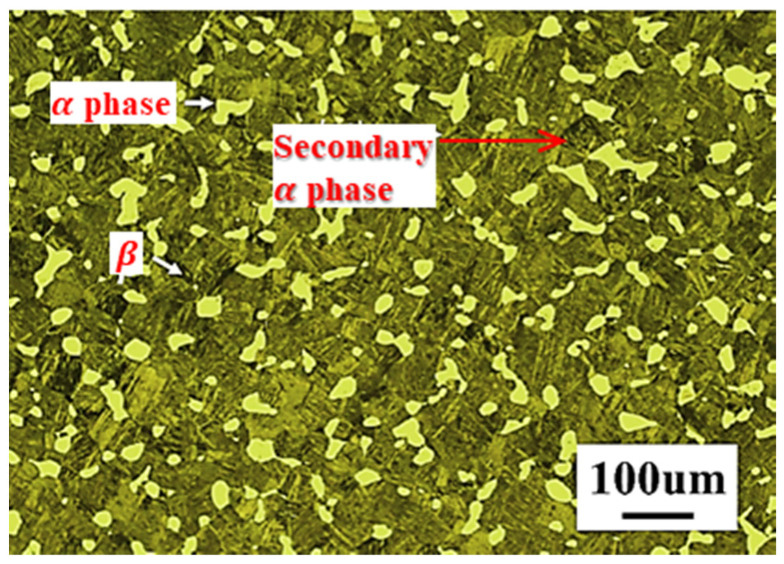
The microstructure of the Ti6242s alloy.

**Figure 2 materials-16-02928-f002:**
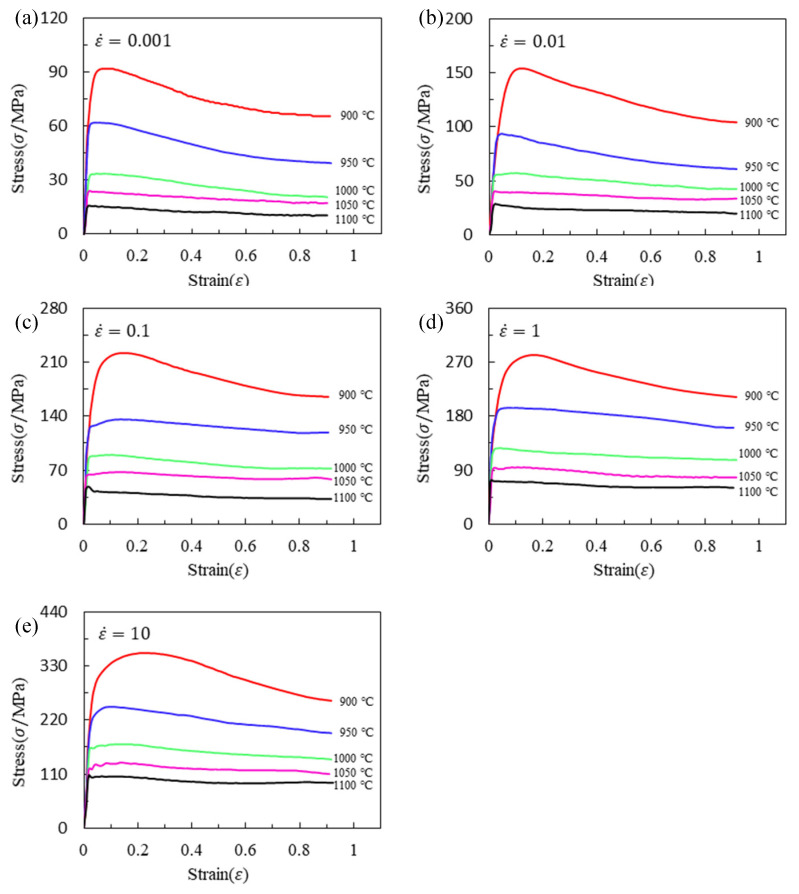
Flow stress–strain curves of Ti6242s alloy during hot compression at strain rates of (**a**) 0.01 s^−1^; (**b**) 0.1 s^−1^; (**c**) 1 s^−1^; (**d**) 10 s^−1^; (**e**) 10 s^−1^.

**Figure 3 materials-16-02928-f003:**
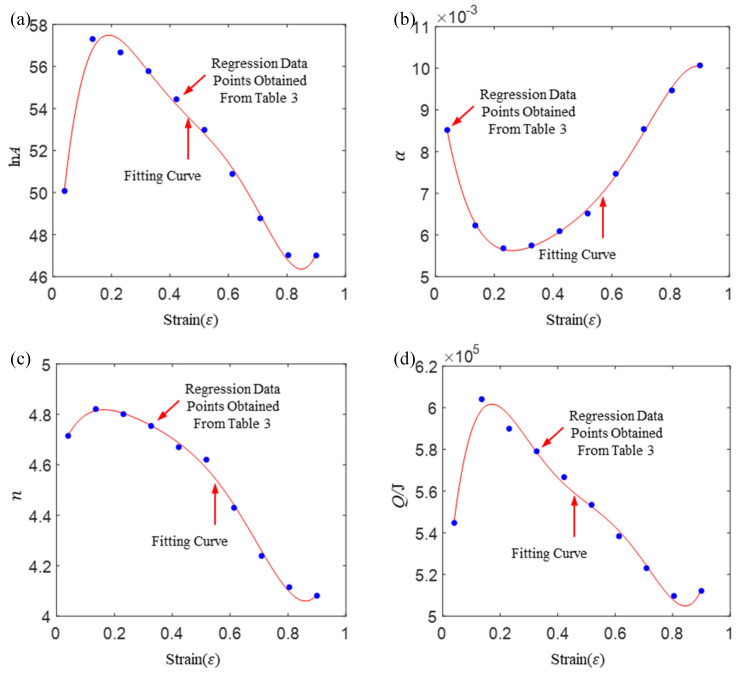
Nonlinear regression values of material parameters at different strain levels and their polynomial fitting: (**a**) lnA-ε; (**b**) α-ε; (**c**) n-ε; (**d**) Q-ε.

**Figure 4 materials-16-02928-f004:**
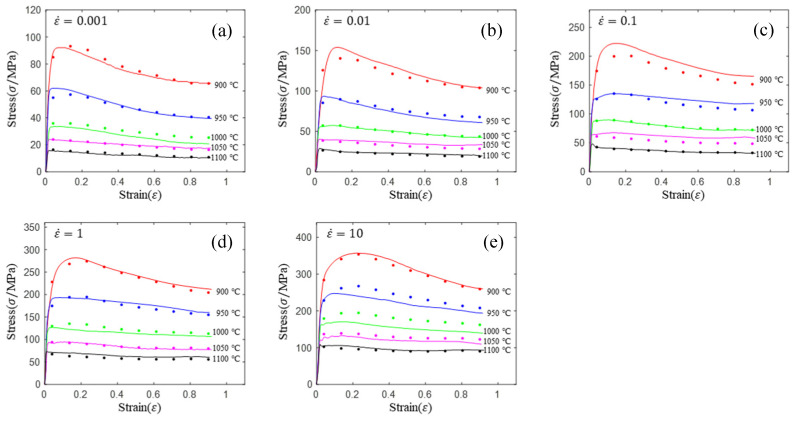
Comparison of prediction flow stress and experimental flow stress for Arrhenius model: (**a**) 0.01 s^−1^; (**b**) 0.1 s^−1^; (**c**) 1 s^−1^; (**d**) 10 s^−1^; (**e**) 10 s^−1^.

**Figure 5 materials-16-02928-f005:**
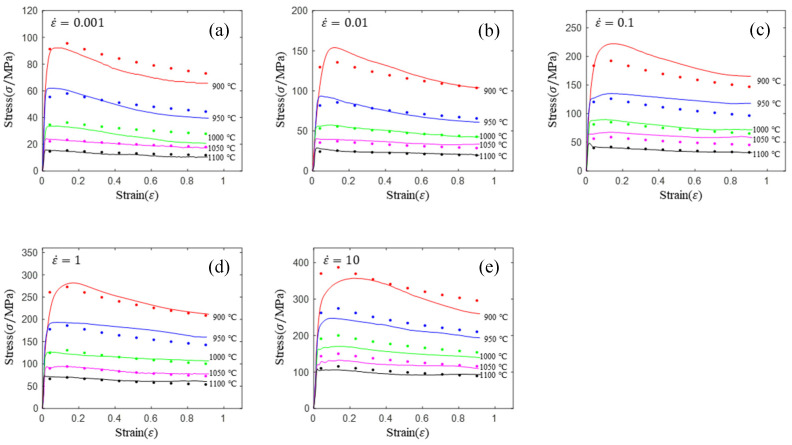
Comparison of prediction flow stress and experimental flow stress for HS model: (**a**) 0.01 s^−1^; (**b**) 0.1 s^−1^; (**c**) 1 s^−1^; (**d**) 10 s^−1^; (**e**) 10 s^−1^.

**Figure 6 materials-16-02928-f006:**
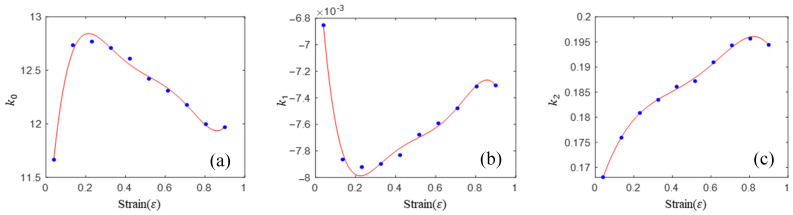
Linear regression values of material parameters at different strain levels and their polynomial fitting: (**a**) Material parameter of $k_{0}$; (**b**) Material parameter of $k_{1}$; (**c**) Material parameter of $k_{2}$.

**Figure 7 materials-16-02928-f007:**
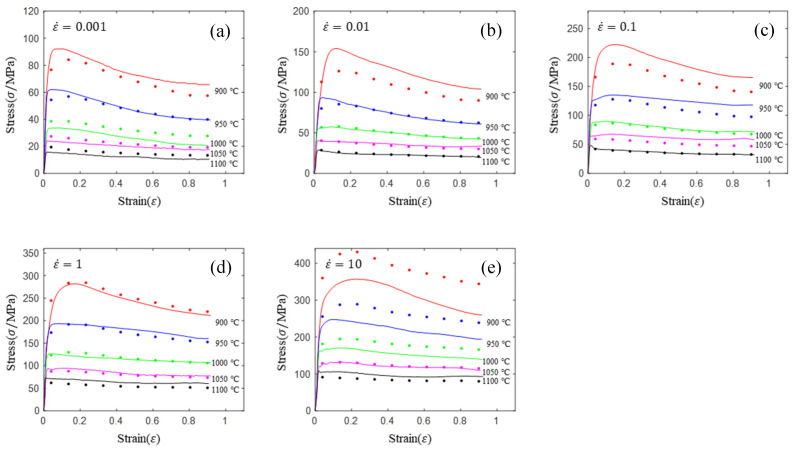
Comparison of prediction flow stress and experimental flow stress for linear model: (**a**) 0.01 s^−1^; (**b**) 0.1 s^−1^; (**c**) 1 s^−1^; (**d**) 10 s^−1^; (**e**) 10 s^−1^.

**Figure 8 materials-16-02928-f008:**
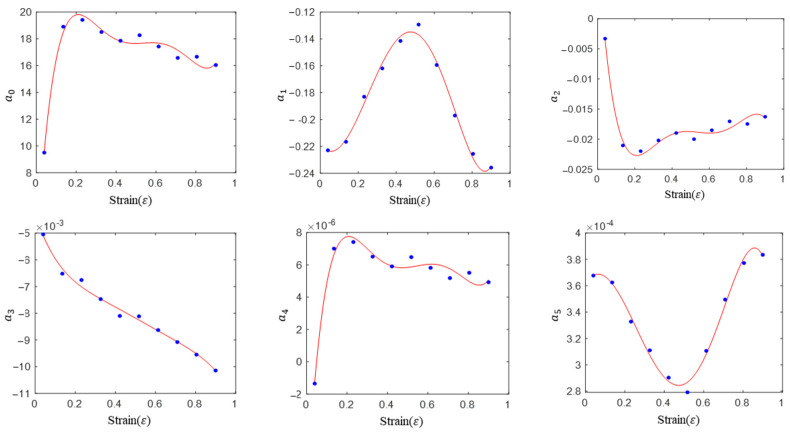
Linear regression values of material parameters at different strain levels and their polynomial fitting for the quadratic model.

**Figure 9 materials-16-02928-f009:**
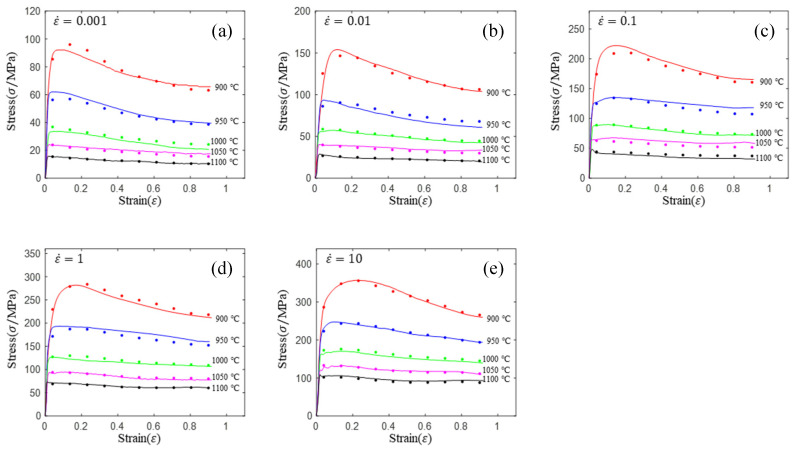
Comparison of prediction flow stress and experimental flow stress for the quadratic model: (**a**) 0.01 s^−1^; (**b**) 0.1 s^−1^; (**c**) 1 s^−1^; (**d**) 10 s^−1^; (**e**) 10 s^−1^.

**Figure 10 materials-16-02928-f010:**
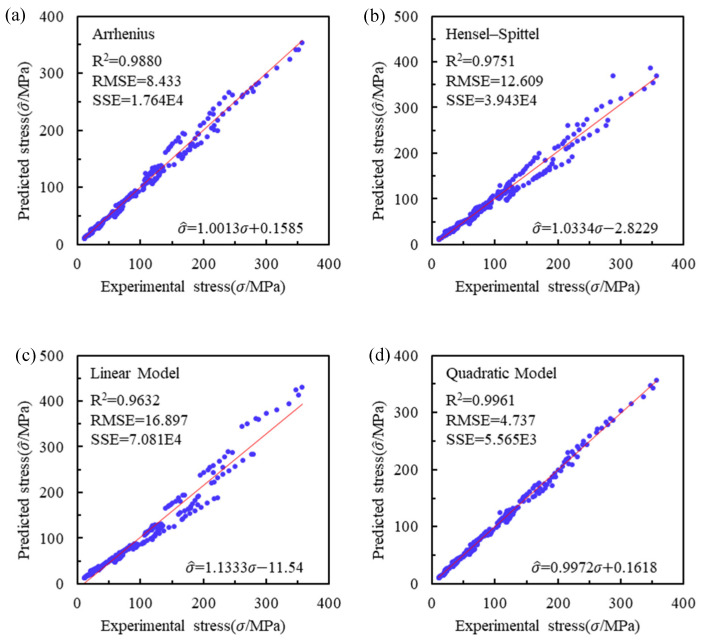
Predictability of flow stress by (**a**) Arrhenius model; (**b**) HS model; (**c**) linear Model; (**d**) quadratic model.

**Figure 11 materials-16-02928-f011:**
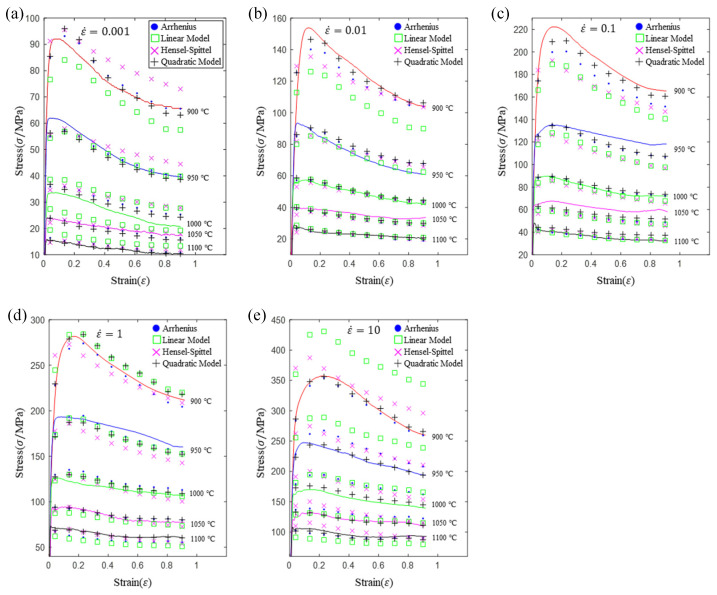
Comparison of prediction flow stress and experimental flow stress for Arrhenius, HS, linear and quadratic models: (**a**) 0.001 s^−1^; (**b**) 0.01 s^−1^; (**c**) 0.1 s^−1^; (**d**) 1 s^−1^; (**e**) 10 s^−1^.

**Figure 12 materials-16-02928-f012:**
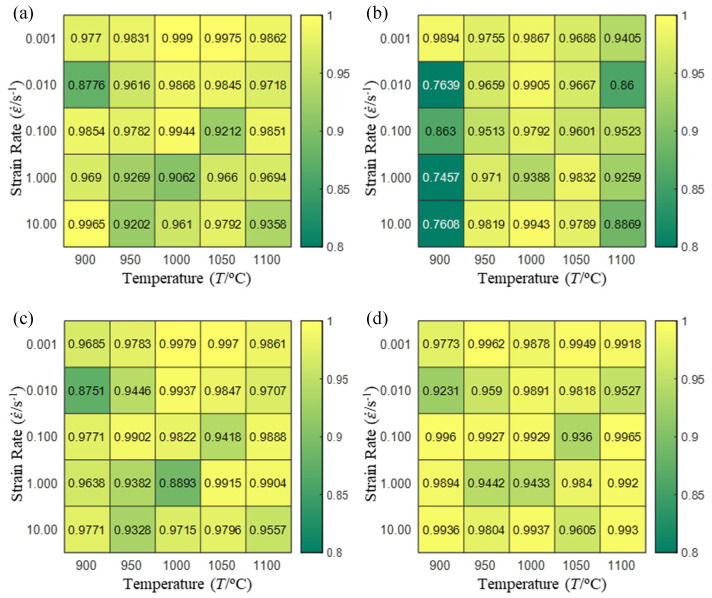
The distribution of the correlation coefficient between predicted flow stress and experimental flow stress on temperature and strain rate: (**a**) Arrhenius model; (**b**) HS model; (**c**) linear model; (**d**) quadratic model.

**Figure 13 materials-16-02928-f013:**
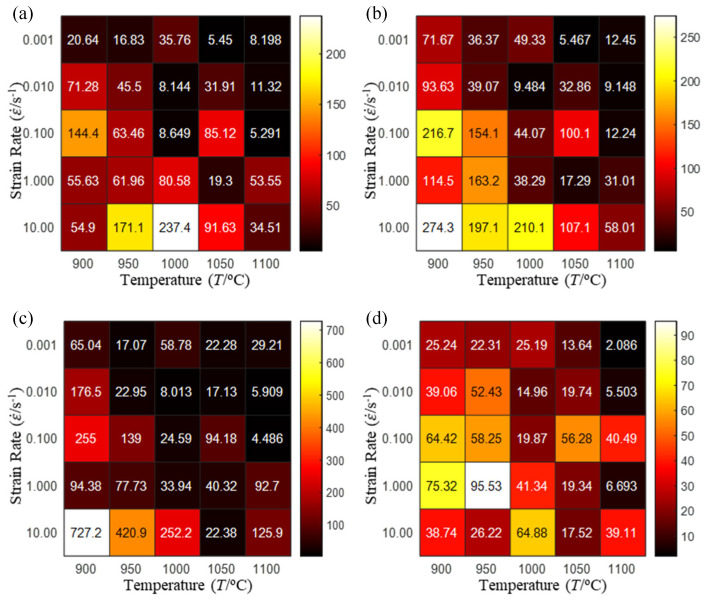
The distribution of the sum of absolute error between predicted flow stress and experimental flow stress on temperature and strain rate: (**a**) Arrhenius constitutive model, (**b**) HS constitutive model, (**c**) linear model, and (**d**) quadratic model.

**Table 1 materials-16-02928-t001:** The normal chemical composition of Ti6242s alloy (wt, %).

Al	Sn	Zr	Mo	Si	Fe	O	N	C
6.13	1.93	4.16	1.90	0.090	0.020	0.09	0.01	0.010

**Table 2 materials-16-02928-t002:** Parameters used in isothermal compression test.

Number	Heating Temperature (T/°C)	Strain Rate ($\dot{\varepsilon }$/s^−1^)	Heating Rate (°C/s)	Quenching Medium	Amount of Deformation
1–5	900	0.001, 0.01, 0.1, 1, 10	5	water	60%
6–10	950				
11–15	1000				
16–20	1050				
21–25	1100				

**Table 3 materials-16-02928-t003:** Stress matrix corresponding to different strain levels (The units of strain rate, temperature and stress are s^−1^, K and MPa, respectively).

		Temperature						Temperature					
ε	$\dot{\varepsilon }$	1173	1223	1273	1323	1373	ε	$\dot{\varepsilon }$	1173	1223	1273	1323	1373
0.04	0.001	88.27	61.94	33.40	23.68	15.63	0.14	0.001	90.86	60.45	32.99	22.93	14.75
	0.01	107.19	93.10	55.79	39.53	27.93		0.01	153.47	89.16	56.73	39.22	24.94
	0.10	180.23	127.29	88.20	64.56	41.71		0.10	222.10	135.27	89.12	67.50	40.31
	1	216.19	190.22	126.78	91.59	71.24		1	279.70	192.35	122.48	94.12	69.67
	10	287.08	230.64	162.64	128.04	104.78		10	347.07	246.11	169.92	132.68	105.52
ε	$\dot{\varepsilon }$	1173	1223	1273	1323	1373	ε	$\dot{\varepsilon }$	1173	1223	1273	1323	1373
0.23	0.001	85.79	56.54	31.49	21.93	13.72	0.33	0.001	80.82	52.74	29.54	21.03	12.68
	0.01	144.71	83.46	53.69	38.27	23.90		0.01	136.61	78.17	51.75	37.34	23.37
	0.10	217.10	133.47	85.32	65.79	38.91		0.10	205.28	130.52	82.28	63.50	37.47
	1	276.49	190.86	118.68	91.53	67.65		1	262.37	186.65	117.12	88.00	65.71
	10	357.03	239.83	166.90	128.26	102.92		10	350.95	233.02	160.30	122.53	97.28
ε	$\dot{\varepsilon }$	1173	1223	1273	1323	1373	ε	$\dot{\varepsilon }$	1173	1223	1273	1323	1373
0.42	0.001	75.54	49.08	27.10	20.07	12.22	0.52	0.001	72.44	45.62	25.59	19.60	12.22
	0.01	130.27	74.16	50.55	36.18	23.11		0.01	122.77	70.14	48.21	34.61	22.85
	0.10	195.80	127.88	79.37	61.76	35.64		0.10	186.98	125.00	75.93	60.19	34.22
	1	250.85	183.04	115.53	84.06	62.75		1	240.83	179.25	112.50	80.59	61.77
	10	336.61	225.79	155.49	120.14	94.07		10	316.86	216.11	151.21	118.28	92.16
ε	$\dot{\varepsilon }$	1173	1223	1273	1323	1373	ε	$\dot{\varepsilon }$	1173	1223	1273	1323	1373
0.61	0.001	69.56	43.27	23.88	18.76	11.31	0.71	0.001	67.54	41.51	22.15	18.27	10.59
	0.01	116.43	66.80	45.77	33.83	22.06		0.01	110.94	64.39	45.04	32.79	21.38
	0.10	178.63	122.68	73.59	58.57	33.41		0.10	171.54	119.85	71.96	58.17	33.00
	1	231.14	175.09	110.75	79.28	61.07		1	222.91	169.64	109.76	78.25	60.79
	10	299.39	210.62	148.26	117.06	91.55		10	283.90	206.85	145.62	116.98	92.55
ε	$\dot{\varepsilon }$	1173	1223	1273	1323	1373	ε	$\dot{\varepsilon }$	1173	1223	1273	1323	1373
0.80	0.001	66.31	40.44	21.25	17.72	10.46	0.90	0.001	65.70	39.61	20.61	17.45	10.42
	0.01	106.63	62.32	42.61	32.78	20.84		0.01	104.12	61.01	42.45	33.38	19.86
	0.10	167.04	117.42	72.37	59.36	33.00		0.10	165.31	118.23	72.00	58.85	32.19
	1	216.97	163.66	108.31	78.01	61.28		1	212.39	160.23	106.84	77.44	60.30
	10	270.08	200.75	143.65	115.29	93.98		10	260.68	194.34	140.07	109.96	92.97

**Table 4 materials-16-02928-t004:** Multiple linear regression results of material parameters.

lnA	$m_{1}$	$m_{2}$	$m_{3}$	$m_{4}$	$m_{5}$	$m_{6}$	$m_{7}$	$m_{8}$
120.3377	0.0069	−0.1544	−0.2426	−0.0135	0.0001	−0.1581	0.0003	−17.4000

**Table 5 materials-16-02928-t005:** The first, second and third partial derivatives of logarithmic stress with respect to temperature at low (ε=0.04), medium (ε=0.23) and high (ε=0.90) strain levels for the experiments’ flow stress curves.

	Order of Partial Derivatives of Logarithmic Stress with Respect to Temperature			
*ε*	lnσ	∂lnσ/∂T	$\partial^{2}\ln\sigma /\partial T^{2}$	$\partial^{3}\ln\sigma /\partial T^{3}$
0.04	(a)	(b)	(c)	(d)
	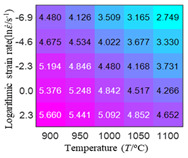	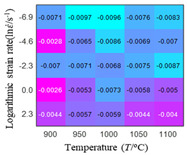	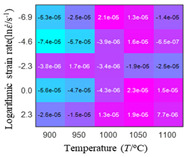	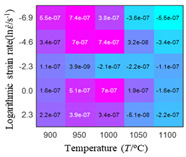
0.23	(e)	(f)	(g)	(h)
	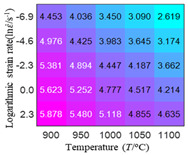	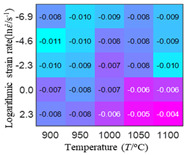	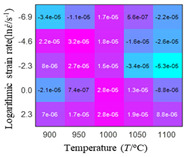	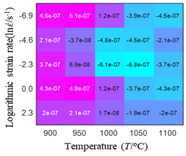
0.9	(i)	(j)	(k)	(l)
	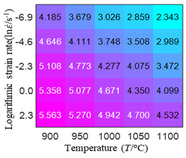	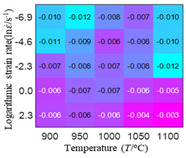	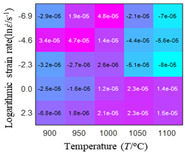	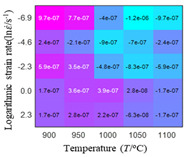

**Table 6 materials-16-02928-t006:** The first, second and third partial derivatives of logarithmic stress with respect to the variable logarithmic strain rate at low (ε=0.04), medium (ε=0.23) and high (ε=0.90) strain levels for experiment flow stress curves.

	Order of Partial Derivative of Logarithmic Stress with Respect to the Variable Logarithmic Strain Rate			
**Strain**	lnσ	$\partial \ln\sigma /\partial \ln\dot{\varepsilon }$	$\partial^{2}\ln\sigma /\partial {\left(\ln\dot{\varepsilon }\right)}^{2}$	$\partial^{3}\ln\sigma /\partial {\left(\ln\dot{\varepsilon }\right)}^{3}$
0.04	(a)	(b)	(c)	(d)
	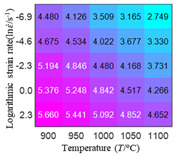	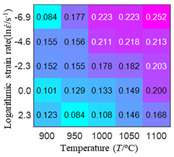	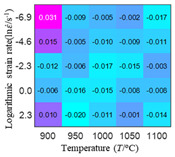	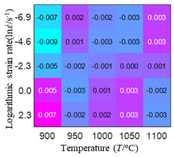
0.23	(e)	(f)	(g)	(h)
	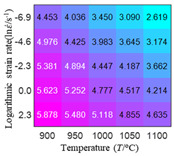	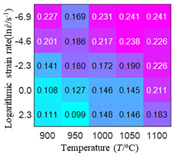	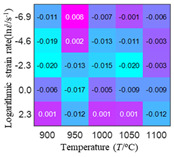	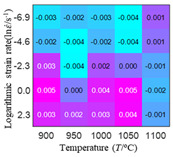
0.9	(i)	(j)	(k)	(l)
	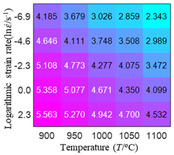	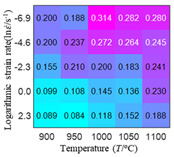	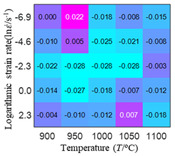	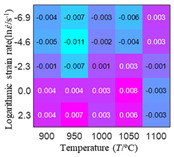

**Table 7 materials-16-02928-t007:** Quantitative comparison of predictability of different models.

Indexs	Arrhenius Model	HS Model	Linear Model	Quadratic Model
R	0.9939	0.9874	0.9814	0.9980
RMSE	8.433	12.609	16.897	4.3737
SSE	17640	39430	70810	5565
SAE (MPa)	1422.52	2097.60	2827.67	884.17

**Table 8 materials-16-02928-t008:** The first, second and third partial derivative of logarithmic stress with respect to temperature for different models at ε=0.04.

.	Order of Partial Derivatives of Logarithmic Stress with Respect to Temperature			
Model	lnσ	∂lnσ/∂T	$\partial^{2}\ln\sigma /\partial T^{2}$	$\partial^{3}\ln\sigma /\partial T^{3}$
Experimental data	(a)	(b)	(c)	(d)
	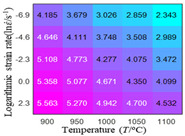	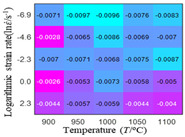	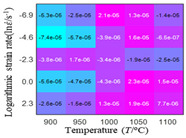	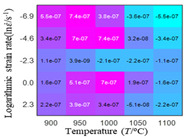
Arrhenius model	(e)	(f)	(g)	(h)
	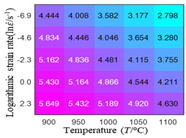	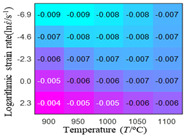	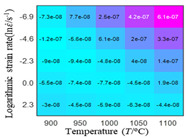	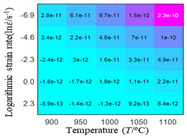
HS Model	(i)	(j)	(k)	(l)
	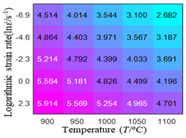	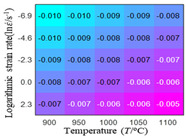	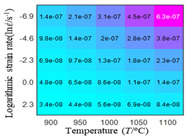	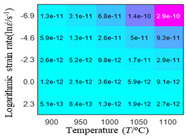
Linear model	(m)	(n)	(o)	(p)
	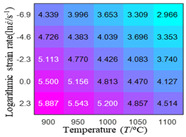	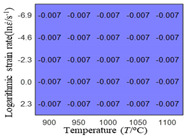	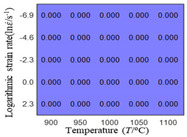	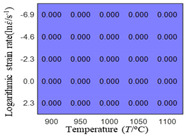
Quadratic model	(q)	(r)	(s)	(t)
	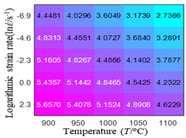	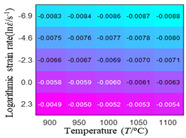	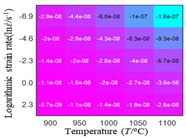	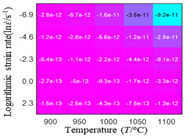

**Table 9 materials-16-02928-t009:** The first, second and third partial derivatives of logarithmic stress with respect to strain rate for different models at ε=0.04.

**.**	Order of Partial Derivatives of Logarithmic Stress with Respect to Temperature			
**Model**	lnσ	$\partial \ln\sigma /\partial \ln\dot{\varepsilon }$	$\partial^{2}\ln\sigma /\partial {\left(\ln\dot{\varepsilon }\right)}^{2}$	$\partial^{3}\ln\sigma /\partial {\left(\ln\dot{\varepsilon }\right)}^{3}$
Experimental data	(a)	(b)	(c)	(d)
	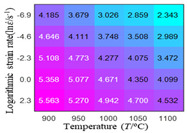	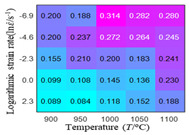	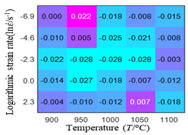	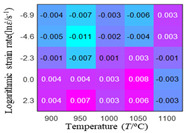
Arrhenius model	(e)	(f)	(g)	(h)
	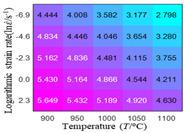	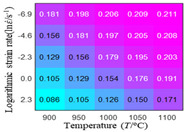	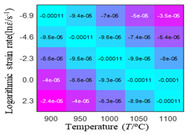	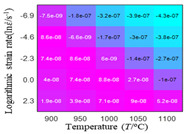
HSModel	(i)	(j)	(k)	(l)
	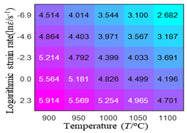	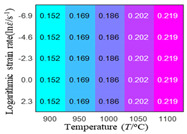	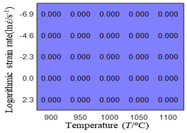	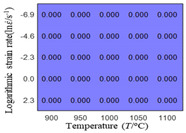
Linear model	(m)	(n)	(o)	(p)
	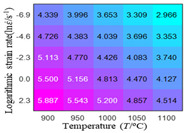	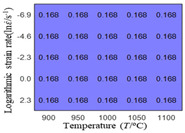	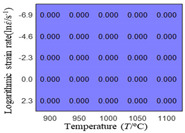	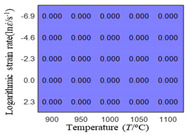
Quadratic model	(q)	(n)	(r)	(s)
	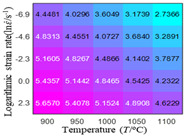	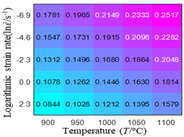	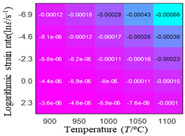	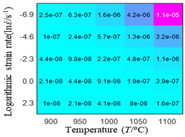

## Data Availability

All data generated or analyzed during this study are included in this published article.

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
