# Peer review of "The Quadratic Constitutive Model Based on Partial Derivative and Taylor Series of Ti6242s Alloy and Predictability Analysis"

_materials, 2023, doi:10.3390/ma16072928_

Round 1
Reviewer 1 Report
In the presented work, an attempt was made to develop a new constitutive model for Ti6242s alloy. The manuscript is well organized and written. The paper is of appropriate length. The title and abstract are satisfactory. I found the approach and conclusions to be robust and useful. The number of bibliographic references is sufficient. The state-of-the-art review presented in the Introduction part is comprehensive and follows a good logical structure. The testing guidelines and equipment used for carrying out the experiments are fully provided, and the obtained results are thoroughly presented and discussed accordingly. Finally, the Conclusions part does a good job in wrapping up the paper by summarizing the main findings. However, here are some comments and suggestions which can help improve the quality of the manuscript.
· The novelty of the work should be highlighted in the manuscript.
· Why was this specific alloy chosen? What are the commercial applications of the material? This can be mentioned in the manuscript.
· The image of the experimental setup needs to be included in the manuscript
· The authors have compared their model with the Arrhenius model and Hensel-Spittel model. How accurate is the model developed by the authors? Authors can provide statistical value to justify the same.
· Quality of the graphs should be improved
· Better quality microstructure image indicating different phases can be included.
· Figures need improvement as some graphs are not clear
· °C is noted shown properly in some places
· Size of the fonts used for the equation is small
· The authors should inform to the audience what is novel in the work carried out.
· The authors are requested to do a thorough proofreading to improve the quality of the manuscript
Reviewer 2 Report
1. While the introduction of the article provides a comprehensive overview of the background and objectives of the study, there are a few limitations that could be added to provide a more balanced perspective. Some potential limitations that could be mentioned include:
a. The introduction could explain why the predictability of classical models for Ti6242s alloy is insufficient and what specific challenges the authors aimed to address with their proposed model. This would help readers understand the context and motivation behind the study.
b. The introduction could provide a brief overview of the results of the study and how well the proposed model performed in comparison to classical models. This would help readers understand the contribution and significance of the study.
2. Text in Fig. 1 is blurry, needs to be corrected. Also, which device was used to take this microstructure, pls show us.
3. Please indicate the basis for choosing High temperature isothermal compression test parameters.
4. Please indicate the basis for selecting 25 cylindrical specimens. Do you think the Design of Experimental is appropriate to use in this case?
5. Pls show us one example of getting the data in Table 3 as you mentioned that can be obtained by linear interpolation using each rheological curve in Figure 2.
6. Classical constitutive model and New constitutive model: Let's briefly summarize some new points built or developed from the Classical model.
7. The main differences among the Quadratic model, Arrhenius model, Linear model and HS model are their order of approximation, the factors they consider (temperature, strain rate, etc.), and the mechanisms they assume to control the deformation of materials. Please show us the reason for choosing these models and the corresponding parameters.
Reviewer 3 Report
There is nothing to critique from my opinion except the title. The paper is well organized and clear written. The title does not clearly reflect the paper content dealing with deformation behaviour.
The paper presents a numerical approach to model the stress – strain relationship that can be used in certain engineering/digital applications. On the other hand the paper does not contribute remarkably to the physics of materials.
Reviewer 4 Report
The authors investigated a suitable constitutive model to predict the high-temperature flow behavior of Ti6242s alloy in both α + β phase and single β phase regions. They found that the prediction accuracy of the Quadratic model is higher than other models. They were able to explain an experimental result of strain provided using this model.
1. What is the main question addressed by the research? At p.2, you wrote that “Because Arrhenius, John-son-Cook (JC) and Hense-Spittle (HS) Model have the characteristics of simple structure and high accuracy, they are currently the mainstream models. However, these models have not made great breakthroughs since they were proposed. Most of the researches are based on these basic models to make some modifications, but these modifications have limited improvement on the accuracy of the model.”
Describe it clearly how you solved this problem that you showed as above.
2. In conjunction with 1., what does it add to the subject area compared with other published articles and materials?
3. p.13 Eq.(11) : You suggested the Q model of the Eq.(11) . Specify which article you thought about this model in reference to.
4. I think the references are suitable for this article. In discussions and results sections, the authors should describe that what kinds of developments were achieved compared with the result of the article of references definitely.
5. Comments for figures and tables ;
- Figure 8, a0, a2, a4: The blue data points were rippling. Is this an essential thing? Show the deviation value of these data points.
- Table8, 9 : Seeing from the right, a title (... model) of the left edge be readable.
- The titles should turn 180 degrees.
